# Carbon Balance of Miscanthus Biomass from Rhizomes and Seedlings

Darija Bilandžija [1,*], Renato Stuparić [2], Marija Galić [1], Željka Zgorelec [1], Josip Leto [3] and Nikola Bilandžija [4]

1 Department of General Agronomy, Division for Agroecology, Faculty of Agriculture, University of Zagreb, Svetošimunska cesta 25, 10000 Zagreb, Croatia; mcacic@agr.hr (M.G.); zzgorelec@agr.hr (Ž.Z.)
2 Graduate Study in Agricultural Engineering–Mechanization, Department of General Agronomy, Division for Agroecology, Faculty of Agriculture, University of Zagreb, Svetošimunska cesta 25, 10000 Zagreb, Croatia; stuparicrenato.p@gmail.com
3 Department of Field Crops, Forage and Grassland, Division for Plant Sciences, Faculty of Agriculture, University of Zagreb, Svetošimunska cesta 25, 10000 Zagreb, Croatia; jleto@agr.hr
4 Department of Agricultural Engineering, Division for Agricultural Engineering and Technology, Faculty of Agriculture, University of Zagreb, Svetošimunska cesta 25, 10000 Zagreb, Croatia; nbilandzija@agr.hr
* Correspondence: dbilandzija@agr.hr; Tel.: +385-12394089

**Abstract:** Biological carbon sequestration is considered an important strategy to mitigate climate change. The energy crop *Miscanthus × giganteus* has great sequestration potential. The objective of this study was to determine: a) the dry matter yield and carbon content in aboveground and belowground biomass; b) the total carbon balance in the plant and soil pool. The study was conducted in continental Croatia (N 45°51′01.32″; E 16°10′35.85″) by the destructive harvesting of five-year-old *Miscanthus × giganteus* stands established by rhizomes (MxgR) and seedlings (MxgS) in the spring of 2021. The soil sampling was conducted in 2016 and 2022. The average amount of carbon in the aboveground biomass of MxgR and MxgS is 11.51 t/ha and 9.87 t/ha, respectively, and in the belowground biomass it is 13.18 t/ha and 14.90 t/ha, respectively. The carbon balance in the plant pool of MxgR is three times lower (1.67 t/ha) than that in the plant pool of MxgS (5.03 t/ha). The total soil carbon content increased by 8.7 t/ha under MxgR and by 14.8 t/ha under MxgS during 2016–2022. Therefore, under the studied agroecological conditions, seedlings should be preferred over rhizomes in the selection of planting material. The obtained data represent valuable input data for sequestration modeling.

**Keywords:** climate change; biological carbon sequestration; *Miscanthus × giganteus*; belowground biomass; aboveground biomass; planting material

## 1. Introduction

Climate change is caused by the emission of greenhouse gasses into the atmosphere from natural and anthropogenic sources. It affects the air temperature; the amount, distribution, and intensity of precipitation; the soil moisture, the sea level, etc. [1]. Since the industrial revolution, human activities have led to a significant increase in greenhouse gas emissions into the atmosphere. In the 19th century, the $CO_2$ concentration was below 250 ppm. However, at the beginning of the 20th century, it started to increase, reaching 419 ppm in 2019 [2]. The global surface temperature will continue to rise until at least mid-century under all of the emission scenarios considered. Global warming is projected to exceed 1.5 °C and 2 °C during the 21st century [3].

$CO_2$ accounts for the largest share of greenhouse gas concentrations, with a share of 80–82% [4]. The sources of $CO_2$ emissions can be natural and anthropogenic. Natural sources of $CO_2$ emissions include, for example, volcanic eruptions (ash); the reduction in forest cover due to extreme weather events such as tornadoes, typhoons, and the shifting of

tectonic plates; and fires. It is assumed that the amount of $CO_2$ that enters the Earth's atmosphere over the course of one year due to volcanic activity is 0.13 Gt [5,6]. Anthropogenic sources of $CO_2$ emissions mainly include the energy, transport, and industrial processes sectors, i.e., activities such as the burning of fossil fuels and the production of textiles, paper, plastics, metals, etc. The agricultural sector contributes 10–20% of the total anthropogenic greenhouse gases, of which carbon dioxide ($CO_2$), methane ($CH_4$), and nitrous oxide ($N_2O$) are the most prominent [1,2,7,8].

The problem of climate change should be solved by long-term solutions, not short-term solutions. Part of a long-term solution in the agricultural sector is the cultivation of alternative renewable energy sources and biological carbon (C) sequestration [9]. Biological carbon sequestration means the uptake of atmospheric carbon through the process of photosynthesis, in which plants absorb some of the atmospheric carbon into their biomass and release another part back into the atmosphere through the process of respiration [10,11]. Biological carbon sequestration can be achieved by changing agricultural management with the aim of increasing the concentration of carbon in plant and soil pools [12].

Energy crops have great potential to meet the world's high energy demand. They also have great sequestration potential. Energy crops are divided into two groups: fast-growing woody plants and perennial rhizomatous grasses. Fast-growing woody plants include, for example, willows and poplars. Perennial rhizomatous grasses include, for example, switchgrass, reed, giant reed, and miscanthus. The advantage of perennial grasses over woody plants is that they form plantations very quickly and produce more biomass at the end of the year, with a very low moisture content [13]. Perennial grasses such as miscanthus are characterized by long periods of exploitation, low cultivation requirements, and the possibility of cultivation on marginal soils [14]. As miscanthus can be grown on marginal soils, it would be suitable for cultivation on set-aside agricultural land worldwide and also in Croatia.

Miscanthus (*Miscanthus × giganteus*) is a highly productive, sterile, rhizomatous perennial grass from Japan [15]. It is a triploid perennial plant with thick and strong rhizomes that form a highly branched root system and a storage site for plant reserve substances [16,17]. Ninety percent of the root system is located in a shallow surface layer up to 35 cm in soil depth, but part of the root system penetrates deeper than 2 m into the soil [16,17].

Miscanthus can be propagated vegetatively by rhizomes and seedlings. The rhizome propagation method is cheaper and allows for the development of stronger and more resistant plants than seedling propagation. The rhizomes must have two to three buds and be stored under cool conditions [18]. The best results in establishing miscanthus stands are obtained when large, healthy rhizomes (about 20 cm long) are planted which have not been stored before planting and are planted at a depth of 20 cm. Studies also show that planting rhizomes results in better growth, a higher dry matter yield, and better overwintering than those of the plants obtained by micropropagation [17]. Tissue culture (in vitro) technology is used to obtain seedlings. The advantage of establishing miscanthus stands with seedlings is the longer time for soil preparation and the removal of perennial weeds [17].

The potential of miscanthus for C storage is considerable, mainly due to the long cultivation period, the large amount of biomass produced, and the low nutrient and water requirements. Miscanthus has been found to store 2.2 t of carbon per ha in one year [19], e.g., 17.7 t/ha in a 16-year-old plantation at a soil depth of 60 cm in Germany [20]. Nevertheless, previous studies have shown that miscanthus may have positive [13,21,22], neutral [9,23], or even negative [22,24] sequestration potential, depending on the specific agroecological conditions.

The increasing interest in miscanthus should be accompanied by research into the site-specific carbon budgets of different planting materials in order to determine the most suitable planting material in view of climate change. In the past, the relationships between biomass formation, i.e., the morphological characteristics and the type of miscanthus planting material, have been identified. Therefore, it was hypothesized that different

planting materials will also lead to different carbon budgets. The novelty of this study lies in the determination of the carbon budgets of miscanthus from different planting materials, which, to our knowledge, has not been studied before, and the data from this study can later be used for sequestration modelling. The objectives of this study were: (1) to determine the dry matter yield and the amount of sequestered carbon in miscanthus biomass (aboveground, belowground, and total biomass) from rhizomes and seedlings; (2) to determine the carbon balances within the plant and soil-plant pool of miscanthus biomass from rhizomes and seedlings.

## 2. Materials and Methods

### 2.1. Experimental Site

The study was conducted at the experimental station "Šašinovec" of the Faculty of Agriculture, University of Zagreb (FAUZ). The site is located in the continental part of northwestern Croatia, near the city of Zagreb (N 45°51′01.32″; E 16°10′35.85″). The experimental field was established in 2016 by planting two types of different planting materials of *Miscanthus* × *giganteus* (Greef et Deu): rhizomes (MxgR) and seedlings (MxgS). The MxgR field has a size of 4 m × 10 m. The rhizomes originate from Croatia and were planted manually at a distance of 1 m between and within the rows at a depth of about 15 cm. The MxgS field is 2.4 m × 10 m, and the seedlings were obtained from Poland and planted with an adapted vegetable planter at a distance of 80 cm between and within the rows. The same agrotechnical measures were applied in both studied treatments.

### 2.2. Climate

The research area has a temperate continental climate. According to the Köppen classification, the study area has a "Cfwbx" climate (a temperate rainfall climate). During 1991–2018, the average annual air temperature was 11.8 °C, the average annual rainfall was 867 mm, the evapotranspiration was 618 mm, and the rainfall factor was 73.3, indicating a semi-humid climate [25].

### 2.3. Soil

Soil sampling to determine the chemical properties of the soil at the experimental site was conducted in 2016 before the field was established. The sampling was conducted in three replicates using the Eijkelkamp soil probe in the top soil layer (0–30 cm soil depth). The soil at the experimental site has alkaline reaction ($pH_{KCl}$ = 7.26); contained 0.11% total nitrogen, 2% total carbon (1.09% total organic carbon and 0.91% total inorganic carbon), and a low carbonate content (7.6%); was well supplied with plant-available potassium (187 mg $K_2O$/kg soil), and was very well supplied with plant-available phosphorus (430 mg $P_2O_5$/kg soil). To determine the changes in the soil carbon content due to miscanthus cultivation, further soil sampling was conducted in 2022 in three replicates using the Eijkelkamp soil probe in the top soil layer (0–30 cm soil depth) in the MxgR and MxgS treatments.

### 2.4. Biomass Sampling

Biomass sampling was conducted during the spring harvest in April 2021 by the destructive harvest of aboveground and belowground biomass. The sampling was conducted on two 5-year-old miscanthus stands established by rhizomes (MxgR) and seedlings (MxgS) in three replicates. The aboveground biomass was destructively harvested from an area of 1 m$^2$ by cutting the plants with a chainsaw (Stihl, Germany) at a height of 10–15 cm from the soil surface. After weighing, the samples were chopped, stored in sampling bags, and transported to the laboratory for further analysis of the plant material. The sampling of the belowground biomass was done by the destructive harvesting of the rhizome and root system at a depth of about 30–35 cm. In addition to the belowground biomass, the stubble was also sampled as part of the belowground biomass. After cleaning the belowground

biomass from the soil particles, the biomass was weighed and taken to the laboratory for further analysis.

*2.5. Laboratory Analysis*

The total carbon content of the aboveground and belowground biomass was determined simultaneously with the dry combustion method. The samples of the plant material were dried to a constant weight in an oven (Nüve, FN 120, Ankara, Turkey) at 105 °C, weighed into tin foils (50 mg $\pm$ 2 mg) (Sartorius CP 64; d = 0.1 mg, Goettingen, Germany), and then analyzed using the Vario Macro CHNS analyzer (Elementar, Langenselbold, Germany). The total carbon content was determined according to the standardized HRN ISO 10694: 2004 protocol.

*2.6. Carbon Balance*

The carbon balance represents the difference between the carbon sink and the carbon source. The carbon sink represents the amount of carbon that remains in the agroecosystem, and the carbon source represents the amount of carbon that is removed from the agroecosystem. In this analysis, following carbon balances were calculated: (1) the carbon balance within the plant pool ($C_{plant}$ balance); (2) the total carbon balance within the soil ($C_{soilTC}$ balance), (3) the total organic carbon balance within the soil pool ($C_{soilTOC}$ balance), and (4) the total inorganic carbon balance within the soil pool ($C_{soilTIC}$ balance):

$$C_{plant} \text{ balance (t/ha)} = C \text{ bgbm (t/ha)} - C \text{ agbm (t/ha)} \tag{1}$$

$$C_{soilTC} \text{ balance (t/ha)} = C \text{ soil}_{TC2022} \text{ (t/ha)} - C \text{ soil}_{TC2016} \text{ (t/ha)} \tag{2}$$

$$C_{soilTOC} \text{ balance (t/ha)} = C \text{ soil}_{TOC22} \text{ (t/ha)} - C \text{ soil}_{TOC16} \text{ (t/ha)} \tag{3}$$

$$C_{soilTIC} \text{ balance (t/ha)} = C \text{ soil}_{TIC22} \text{ (t/ha)} - C \text{ soil}_{TIC16} \text{ (t/ha)} \tag{4}$$

where:
C agbm—carbon content in the aboveground biomass (t/ha),
C bgbm—carbon content in the belowground biomass (t/ha),
C soil$_{TC16}$—total soil carbon content in 2016 (t/ha),
C soil$_{TC22}$—total soil carbon content in 2022 (t/ha),
C soil$_{TOC16}$—total soil organic carbon content in 2016 (t/ha),
C soil$_{TOC22}$—total soil organic carbon content in 2022 (t/ha),
C soil$_{TIC16}$—total soil inorganic carbon content in 2016 (t/ha),
C soil$_{TIC22}$—total soil inorganic carbon content in 2022 (t/ha),

*2.7. Statistical Analysis*

The statistical analysis was performed using SAS 9.1 statistical software (SAS Inst. Inc., 2002–2004, Cary, NC, USA). Variability among the studied planting materials was analyzed using an analysis of variance (ANOVA) and, if necessary, tested with a post-hoc (Fisher) *t*-test. The significance threshold for all the analyses was 5%. The prescribed quality control procedures were carried out in the analytical laboratory of the Department of General Agronomy of FAUZ.

**3. Results**

*3.1. Dry Matter Yield and Carbon Content in Miscanthus Biomass*

The analysis of variance revealed no statistically significant difference between the studied planting materials for the dry matter and carbon content in the aboveground ($p = 0.0551$; $p = 0.1115$), belowground ($p = 0.2154$; $p = 0.1962$), and total ($p = 0.447$; $p = 0.4366$) miscanthus biomass, respectively.

The average dry matter yields of the MxgR and MxgS aboveground biomass were 25.86 t/ha and 22.04 t/ha, respectively, and those of the belowground biomass were

29.44 t/ha and 33.43 t/ha, respectively (Table 1). As for the total dry matter yield, the average was 55.30 t/ha for MxgR and 55.47 t/ha for MxgS.

**Table 1.** Dry matter yield and carbon content in miscanthus biomass.

| Planting Material | DM Yield (t/ha) | C Content (%) | C Content (t/ha) |
|---|---|---|---|
| **Aboveground biomass** | | | |
| MxgR | 25.86 ($\pm$5.45) | 44.49 ($\pm$1.64) | 11.51 ($\pm$7.10) A |
| MxgS | 22.04 ($\pm$4.75) | 44.79 ($\pm$0.54) | 9.87 ($\pm$5.27) A |
| **Belowground biomass** | | | |
| MxgR | 29.44 ($\pm$10.62) | 44.78 ($\pm$0.81) | 13.18 ($\pm$10.35) A |
| MxgS | 33.43 ($\pm$10.49) | 44.63 ($\pm$1.53) | 14.90 ($\pm$9.14) A |
| **Total biomass** | | | |
| MxgR | 55.30 ($\pm$7.52) | 44.63 ($\pm$1.03) | 24.69 ($\pm$8.07) A |
| MxgS | 55.47 ($\pm$7.45) | 44.71 ($\pm$0.68) | 24.78 ($\pm$6.67) A |

Average values marked with the same letters are not statistically significantly different at $p \leq 0.05$; (DM—dry matter yield; $\pm$rsd—relative standard deviation).

The average carbon content in the aboveground biomass of MxgR and MxgS was 44.49% and 44.79%, respectively (Table 1). Thus, the average carbon content in the aboveground biomass of MxgR and MxgS was 11.51 t/ha and 9.87 t/ha, respectively. In the belowground biomass, the average carbon content in MxgR and MxgS was 44.78% and 44.63%, i.e., 13.18 t/ha and 14.90 t/ha, respectively (Table 1). The average carbon content of the total MxgR and MxgS biomass was 44.63% and 44.71%, i.e., 24.69 t/ha and 24.78 t/ha, respectively (Table 1).

*3.2. Carbon Balance*

The statistical analysis of variance showed a statistically significant difference ($p = 0.0226$) in the carbon balance of the plant pool between the studied plant materials. The carbon balance within the plant pool was positive for both MxgR and MxgS and amounted 1.67 t/ha and 5.03 t/ha, respectively. Thus, the carbon balance of MxgS was three times higher than that of MxgR (Table 2). The difference in the morphological development of the Miscanthus plants grown by rhizome division and micropropagation (seedlings) contributed to the significant differences in the carbon balances. Although no statistically significant differences were found for the carbon content in the aboveground and belowground biomass between the studied planting materials, MxgS had a higher carbon content in the belowground biomass and a lower carbon content in the aboveground biomass compared to MxgR. This different distribution patterns of carbon in the aboveground and belowground biomass had a significant impact on the overall carbon balance.

**Table 2.** Carbon balance within the plant pool in relation to the different planting materials.

| Planting Material | C Sink (t/ha) | C Source (t/ha) | C Balance (t/ha) |
|---|---|---|---|
| MxgR | 13.18 | 11.51 | 1.67 A |
| MxgS | 14.90 | 9.87 | 5.03 B |

Average values marked with the same letters are not statistically significantly different at $p \leq 0.05$.

Considering the carbon content in the soil pool, in 2016, the TOC content was 49.7 t/ha and the TIC content was 41.5 t/ha, i.e., the TC content in the soil pool was 91.2 t/ha. The soil TC and TOC content increased, while the TIC content decreased during the study period. In 2022, 64.6 and 69.9 t/ha of TOC and 35.3 and 36.1 t/ha of TIC (i.e., 99.9 and 106.0 t/ha of TC), respectively, were found in the soil pool under MxgR and MxgS (Table 3).

**Table 3.** Carbon balance within the soil pool in relation to the total, organic, and inorganic carbon content.

| TC (t/ha) | | | |
|---|---|---|---|
| **Planting Material** | **C soil$_{TC16}$** | **C soil$_{TC22}$** | **C soil$_{TC}$ balance** |
| **MxgR** | 91.2 | 99.9 | 8.7 A |
| **MxgS** | 91.2 | 106.0 | 14.8 B |
| **TOC (t/ha)** | | | |
| **Planting Material** | **C soil$_{TOC16}$** | **C soil$_{TOC22}$** | **C soil$_{TOC}$ balance** |
| **MxgR** | 49.7 | 64.6 | 14.9 A |
| **MxgS** | 49.7 | 69.9 | 20.2 A |
| **TIC (t/ha)** | | | |
| **Planting Material** | **C soil$_{TIC16}$** | **C soil$_{TIC22}$** | **C soil$_{TIC}$ balance** |
| **MxgR** | 41.5 | 35.3 | −6.2 A |
| **MxgS** | 41.5 | 36.1 | −5.4 A |

Average values marked with the same letters are not statistically significantly different at $p \leq 0.05$.

The statistical analysis of variance revealed a statistically significant difference ($p = 0.0075$) in the TC balances among the studied plant materials. The TC balance within the soil pool was positive for both MxgR and MxgS and was 8.7 t/ha and 14.8 t/ha, respectively (Table 3). In addition, the statistical analysis of variance revealed no statistically significant difference in the TOC ($p = 0.0964$) and TIC ($p = 0.7260$) balances among the plant materials studied. The TOC balance within the soil pool was positive for both MxgR and MxgS and was 14.9 t/ha and 20.2 t/ha, respectively (Table 3). The TIC balance within the soil pool was negative for both MxgR and MxgS and was −6.2 t/ha and −5.4 t/ha, respectively (Table 3).

## 4. Discussion

Not many studies have been conducted to compare the dry matter yield, carbon content, and carbon balance of miscanthus obtained from different planting materials. Similar to this study, Bilandžija et al. [26] determined an average dry matter yield of 25.84 t/ha for aboveground biomass in a study conducted under similar agroecological conditions. Many authors [13,27–31] found lower average dry matter yields—ranging from 6.2 to 26 t/ha—of aboveground biomass. The literature review also shows higher dry matter yields in aboveground biomass compared to the results obtained in this study [32,33]. The average dry matter yield of the aboveground biomass was 38.1 t/ha [32] and 29.6 t/ha [33] for the winter harvest conducted in December. The higher dry matter yields can be explained by the different agroecological conditions and harvest dates, as the yield level decreased in the winter due to senescence and the falling of leaves.

The average carbon content in the aboveground biomass was similar for the different planting materials. The average carbon content in the aboveground biomass of MxgR and MxgS was 44.49% and 44.79%, i.e., 11.51 t/ha and 9.87 t/ha, respectively. Under different agroecological conditions considering climate and soil, the average carbon contents were in the range of 43.59%–49.2% [34–36]. A higher average carbon content of 14.3 t/ha was determined in a four-year-old stand [37]. The results are in accordance with the average carbon content determined under similar agroecological conditions in Croatia. It was found that the average carbon content was 48.59% in the autumn harvest and 49.49% in the spring harvest [26]. However, in another study which was also conducted under agroecological conditions in Croatia, a higher average carbon content (15.7 t/ha) was found in the aboveground biomass due to differences in the annual precipitation distribution and soil moisture availability.

The importance of storing organic C in soils through belowground biomass has been recognized as an important mitigation measure—especially for miscanthus, which allocates a significant amount of carbon in belowground biomass. In this study, the average carbon content in the belowground biomass of MxgR and MxgS was 44.78%, i.e., 13.18 t/ha, and 44.63%, i.e., 14.90 t/ha, respectively. A wide range of dry matter yields and carbon contents were obtained in previous studies [13,28,31,32,37–42]. The present results on the dry matter yield and carbon content in belowground biomass are lower compared to those from some studies [22,31,32,38,39,42,43].

In the March harvest in Germany, a dry matter yield of 11.5 t/ha [38] was determined. Similar results (11.4 t/ha) were obtained in a fourteen-year study [39]. In a study conducted under agroecological conditions in Europe and the USA, a range of 6.1–13 t/ha dry matter yield in belowground biomass was determined [22,31,32,40–42]. Higher dry matter yields in belowground biomass, varying from 11–25 t/ha, were determined in Europe [13,27,31,41]. Even higher dry matter contents were determined in the USA—27.1 t/ha to a depth of 25 cm [32], i.e., 36.8 t/ha—in an irrigated Miscanthus plantation [43]. Similar to this study, a dry matter yield of 16 t/ha in belowground biomass at the harvest in April, which decreased to 13 t/ha by the harvest in August, was determined in a study conducted in Germany [28].

Our sampling method of belowground biomass included the sampling of stubble, total rhizome mass, and only roots near the rhizomes to a depth of 30–35 cm, rather than the entire root system. It is considered that 28% [38], or nearly 90%, of the total Miscanthus root biomass is located at soil depths up to ~35 cm [16]. Therefore, we underestimated the belowground biomass. The carbon storage in the belowground biomass is likely much higher than that which we have directly quantified. It is assumed that the part of the belowground biomass, i.e., the part of the root system, will die each year and regrow with the growth dynamics of the rhizome. Nevertheless, it is very difficult to quantify roots and their recycling in soil [38].

The results of this study show a significant difference in the carbon balance of miscanthus obtained from rhizomes and seedlings. Both of the determined balances are positive and contribute to climate change mitigation. However, the carbon balance of MxgR is three times lower than that of MxgS. Similar results were obtained for the total carbon balance in this study. The determined carbon balances were also positive and ranged from 0.78 to 4.5 t/ha annually [21,22,24,44,45]. The carbon balance is significantly affected by soil properties such as pH, clay content, and carbon content, as well as agrotechnical measures such as tillage, as they affect the rate of decomposition and the partitioning of the SOM residues from each SOM pool at any time [21]. In addition, it is also significantly affected by the climate; it is mainly influenced by different precipitation distributions, the number of sunny days, the temperature, and the solar radiation [24,45].

A significant difference in the carbon balances determined in this study was found due to the different dry matter yields and distribution patterns (ratio) of the aboveground and belowground biomass carbon content in the total miscanthus biomass. The MxgR biomass has 47% of the dry matter yield and carbon content in the aboveground biomass and 53% in the belowground biomass. The MxgS biomass has 40% of the dry matter yield and carbon content in the aboveground biomass and 60% in the belowground biomass. Therefore, the rhizome planting material contributed more root system development and a higher carbon content in the belowground biomass compared to the seedlings. The seedling planting material contributed to higher dry matter yields of the aboveground biomass and a higher carbon content in that biomass than the rhizome planting material did. The different distribution pattern of the aboveground and belowground biomass in the total plant biomass and, consequently, the different total carbon balances can be explained by the different structural and morphological characteristics as well as the different distribution of nutrients between the studied planting materials.

Lewandowski [46], in a study comparing miscanthus obtained from rhizomes and seedlings, found a significant difference in the morphological development of miscanthus.

The observed difference manifested itself in the thickness of the branches and in the number and strength of the shoots. The plants propagated by rhizomes were of a lower number but had stronger and thicker shoots and lower yields compared to the plants obtained from seedlings [46,47].

## 5. Conclusions

This study, conducted in 5-year-old miscanthus stands (to a soil depth of 30 cm), revealed no significant difference in the dry matter yield and carbon content in aboveground and belowground biomass between the studied planting materials (rhizomes and seedlings) and a statistically significant difference in the total carbon balance between the studied planting materials. A different distribution pattern of the dry matter yield and carbon content of the aboveground and belowground biomass in the total plant biomass was observed. In the MxgR biomass, 47% of the dry matter yield and carbon content was in the aboveground biomass, and 53% was in the belowground biomass. The MxgS biomass had 40% of the dry matter yield and carbon content in the aboveground biomass and 60% in the belowground biomass. The carbon balance of the Miscanthus from rhizomes was three times lower (1.67 t/ha) than that of seedlings (5.03 t/ha). The soil total carbon and total organic carbon content increased, while the total inorganic carbon content decreased during the study period in both the MxgR and MxgS treatments. The total carbon and total organic carbon balances of the MxgR (8.7 and 14.9 t/ha, respectively) and MxgS (14.8 and 20.2 t/ha, respectively) treatments were positive and contributed to long-term climate change mitigation. However, both balances of the rhizomes were lower than those of the seedlings. Therefore, under the studied agroecological conditions, seedlings should be preferred over rhizomes in planting, i.e., the selection of planting materials, regardless of the higher investment costs for the seedlings compared to the rhizome production.

**Author Contributions:** Conceptualization, D.B., N.B. and J.L.; methodology, D.B. and N.B.; software, Ž.Z.; validation, Ž.Z.; formal analysis, R.S., D.B., N.B., J.L. and M.G.; investigation, R.S., J.L. and N.B.; data curation, Ž.Z.; writing—original draft preparation, D.B. and R.S.; writing—review and editing, D.B., R.S., N.B., Ž.Z. and J.L.; visualization, R.S.; supervision, D.B. and N.B. All authors have read and agreed to the published version of the manuscript.

**Funding:** The publication was supported by the Open Access Publication Fund of the University of Zagreb Faculty of Agriculture.

**Data Availability Statement:** Not applicable.

**Conflicts of Interest:** The authors declare no conflict of interest.

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
