# Peer review of "Carbon Balance of Miscanthus Biomass from Rhizomes and Seedlings"

_agronomy, doi:10.3390/agronomy12061426_

Round 1

Reviewer 1 Report

The objective of this study was to determine: a) dry matter yield and carbon content in above and below ground biomass b) total carbon balance in the plant and soil pool.

Thematically the work is interesting for the researchers and professionals and the proposed manuscript is relevant to the scope of the journal.

I found it appropriate for publication in the Agronomy journal, but only after some modifications and clarification from the Authors.

The title is a clear representation of the manuscript's content. The abstract reflects realistically the substance of the work.

The overall organization and structure of the manuscript are appropriate. The paper is well written and the topic is appropriate for the journal.
The aim of the paper is well described and the discussion was well approached, its results and discussion are correlated to the cited literature data.

The literature review is comprehensive and properly done.

The novelty of the work must be more clearly demonstrated.

The significance of the Work: Given the large number of analyzed data, this is an interesting study with a possible significant impact in this area.

Other Specific Comments: The work is properly presented in terms of the language. The work presented here is very interesting and well done, it is presented in a compact manner.

In general, there are no doubtful or controversial arguments in the manuscript. The methodology applied in the research is presented in clear manner, so that it is repeatable by other authors.

Please include more numerical values in the Conclusion section,

Author Response

Reviewer - comments:

Please include more numerical values in the Conclusion section

Answer:

Dear reviewer, thank you very much on your constructive comments that have contributed to the improvement of the paper. Numerical values have been included in the Conclusion section.

Kind regards,

Authors

Reviewer 2 Report

The authors have improved the manuscript and have responded appropriately to my concerns. The manuscript is clearly written. The following issues should be fixed.

1. Line 213, the TOC content was 41.5 t/ha TIC?

2. Line 284-286, Carbon balance is significantly affected by soil properties such as pH, clay content, and carbon content, as well as agrotechnical measures such as tillage. The authors did not measured these soil properties, so please add the references.

3. Line 320, Total soil total carbon?

Author Response

Reviewer - comments:

The authors have improved the manuscript and have responded appropriately to my concerns. The manuscript is clearly written. The following issues should be fixed.

  1. Line 213, the TOC content was 41.5 t/ha TIC?
  2. Line 284-286, Carbon balance is significantly affected by soil properties such as pH, clay content, and carbon content, as well as agrotechnical measures such as tillage. The authors did not measured these soil properties, so please add the references.
  3. Line 320, Total soil total carbon?

Answer:

Dear reviewer, thank you very much on your constructive comments that have contributed to the improvement of the paper. The authors have addressed all comments:

  1. Line 213, the TOC content was 41.5 t/ha TIC?

                Changed in TIC content was 41.5 t/ha

  1. Line 284-286, Carbon balance is significantly affected by soil properties such as pH, clay content, and carbon content, as well as agrotechnical measures such as tillage. The authors did not measured these soil properties, so please add the references.

                Reference added

  1. Line 320, Total soil total carbon?

                Changed to Soil total carbon

Kind regards,

Authors

This manuscript is a resubmission of an earlier submission. The following is a list of the peer review reports and author responses from that submission.

Round 1

Reviewer 1 Report

A. General comments: 1) The title of the paper is misleading. There is no sequestration potential methodology, measurements and results as  indicated in objective 3 and also copiously in the text. The title must rather dwell on carbon balance. 2) The article focuses on Miscanthus sequestration potential but virtually nothing (data or analysis) shows that. 3) The paper does not compare Miscanthus with any similar vegetation for the set objectives. 4) English language and grammar must be generally improved. The authors must write and present their data and information in the past tense to be more precise and clear. 5) I can't see the methodology section of the paper. 6) There are some controversial statements which must be addressed.

B. Specific comments: 1) Several grammar and spellings, e.g., Line 14: "One of the possibilities" .....Line 15: "The aim of the study.... "2) Line 16: "for 3 scenarios...." where are the scenarios? Line 26-27: "Results show .....There are no results that show that. It was authors' opinion expressed in line 66-67. 3) Line 32: Sentence is controversial and misleading. Climate change is not only caused by anthropogenic greenhouse emissions. How about natural greenhouse gases from bogs, wetlands, etc.? 4) Writing style is boring and unacceptable in science writing which depends on short sentences and easy comprehension. There are no paragraphs from line 32 until line 100 (one and a half pages long?). 5) Results and Discussion sections are too confusing to read and comprehend. 6) Authors must avoid undefined acronyms, e.g., "Cfwbx"; what is that? 7) The ANOVA (Table 1) seems to be related to Table 2 that compares Miscanthus biomass rhizomes with seedlings and does not need to stand alone. The ANOVA is meaningless by itself. The same problem relates with Tables 3 and 4. 8) The discussion is very long and parameters with other similar trees either in Croatia or other countries. The section must be broken into sections to clarify discussion. 9) The phrase: "According to literature review", e.g., lines 233, 252, 277, etc. is repetitive and unnecessary in science writing and must be removed.

Author Response

Reviewer1 - A. General comments:

1) The title of the paper is misleading. There is no sequestration potential methodology, measurements and results as  indicated in objective 3 and also copiously in the text. The title must rather dwell on carbon balance. 2) The article focuses on Miscanthus sequestration potential but virtually nothing (data or analysis) shows that. 3) The paper does not compare Miscanthus with any similar vegetation for the set objectives. 4) English language and grammar must be generally improved. The authors must write and present their data and information in the past tense to be more precise and clear. 5) I can't see the methodology section of the paper. 6) There are some controversial statements which must be addressed.

Answer:

Dear reviewer, thank you very much on your constructive comments. The title has been changed, results are compared with similar research on miscanthus as paper is focused on it. The methodology section is described in section 2. Materials and Methods, devided on subsections. Language and grammer are improved.

Reviewer1 - B. Specific comments:

1) Several grammar and spellings, e.g., Line 14: "One of the possibilities" .....Line 15: "The aim of the study.... "

Answer: improved

2) Line 16: "for 3 scenarios...." where are the scenarios?

Answer: The scenarios are presented in section 2. Matherials and methods: 2.8. Land Use Change from Abandoned Agricultural Land to Miscanthus Cultivation. They were not described in Abstract sections as the number of words is limited.

Line 26-27: "Results show .....There are no results that show that. It was authors' opinion expressed in line 66-67.

Answer: changed

3) Line 32: Sentence is controversial and misleading. Climate change is not only caused by anthropogenic greenhouse emissions. How about natural greenhouse gases from bogs, wetlands, etc.?

Answer: changed

4) Writing style is boring and unacceptable in science writing which depends on short sentences and easy comprehension. There are no paragraphs from line 32 until line 100 (one and a half pages long?).

Answer: simplified, devided into more paragraphs

5) Results and Discussion sections are too confusing to read and comprehend.

Answer: simplified, devided into more paragraphs

6) Authors must avoid undefined acronyms, e.g., "Cfwbx"; what is that?

Answer: inserted the interpretation of clasification

7) The ANOVA (Table 1) seems to be related to Table 2 that compares Miscanthus biomass rhizomes with seedlings and does not need to stand alone. The ANOVA is meaningless by itself. The same problem relates with Tables 3 and 4.

Answer: deleted

8) The discussion is very long and parameters with other similar trees either in Croatia or other countries. The section must be broken into sections to clarify discussion.

Answer:done, devided into more paragraphs

9) The phrase: "According to literature review", e.g., lines 233, 252, 277, etc. is repetitive and unnecessary in science writing and must be removed.

Answer: removed

Reviewer 2 Report

Authors:

I have inserted comment as "insert text at curser" highlighted in pink color.

  1. The materials and methods section,
  2. tables, and
  3. the conclusion part need significant improvement. Please go through the manuscript and made edits to the comments. 

The reviewer

Author Response

Reviewer - General comments:

I have inserted comment as "insert text at curser" highlighted in pink color.

The materials and methods section, tables, and the conclusion part need significant improvement. Please go through the manuscript and made edits to the comments.

Answer:

Dear reviewer, thank you very much on your constructive comments. The text has been changed and I belive that your comments have contributed to the improvement of the paper

Reviewer - Specific comments:

  1. Line 72

Answer: done,

  1. Line 105

Answer: deleted

  1. Line 132-150

Answer: done

  1. Line 178

Answer: yes, but deleted as it was so required

  1. Line 190

Answer: done

  1. Line 213

Answer: done

  1. Line 312

Answer: done

Reviewer 3 Report

The authors investigate the carbon balance and sequestration potential of Miscanthus biomass obtained from rhizomes and seedlings.The results of this paper may make valuable contributions to the carbon balance for energy crop under global climate change. The topic of the study fits the scope of the journal. In general, the article is well organized, the research method is reasonable, and I do hope that it could make a valuable contribution to the later energy crop planting. However, writing/clarity need additional attention. Moreover, the authors should emphasize the novel of the paper. The conclusion should be modified more carefully. Please provide the related suggestion for energy crop planting and management based on the results of this paper in the conclusion section.

Detail comments:

  1. Introduction section should be divided 2-3 paragraphs.
  2. Please provide hypothesis at the end of introduction.
  3. Line 58, carbon (C), the abbreviated C should be given when it firstly appeared.
  4. Line 117, please add “are”after evapotranspiration amounts.
  5. Pay attention to the digital numbers in table 1.
  6. Please provide the full name of Cv in table 1.
  7. In the reference list, the ordernumber of the fist four references were repeated.
  8. Please pay attention to the uppercase and lowercase lettersin the reference titles.

Author Response

Reviewer1 - A. General comments:

The authors investigate the carbon balance and sequestration potential of Miscanthus biomass obtained from rhizomes and seedlings.The results of this paper may make valuable contributions to the carbon balance for energy crop under global climate change. The topic of the study fits the scope of the journal. In general, the article is well organized, the research method is reasonable, and I do hope that it could make a valuable contribution to the later energy crop planting. However, writing/clarity need additional attention. Moreover, the authors should emphasize the novel of the paper. The conclusion should be modified more carefully. Please provide the related suggestion for energy crop planting and management based on the results of this paper in the conclusion section.

Answer:

Dear reviewer, thank you very much on your constructive comments. The text has been changed and simplified and I belive that it has contributed to the clarity of the paper, the novel of the paper has been emphasized and conclusion has been modified

Reviewer1 - B. Specific comments:

  1. Introduction section should be divided 2-3 paragraphs.

Answer: done,

  1. Please provide hypothesis at the end of introduction.

Answer: done,

  1. Line 58, carbon (C), the abbreviated C should be given when it firstly appeared.

Answer: done,

  1. Line 117, please add “are”after evapotranspiration amounts.

Answer: done,

  1. Pay attention to the digital numbers in Table 1.

Answer: not presented any more as it was required to be deleted

  1. Please provide the full name of Cv in Table 1.

Answer: not presented any more as it was required to be deleted

  1. In the reference list, the ordernumber of the fist four references were repeated.

Answer: changed

  1. Please pay attention to the uppercase and lowercase lettersin the reference titles.

Answer: changed

Round 2

Reviewer 1 Report

The authors have addressed most of this reviewers previous comments in a satisfactory way. In my previous review, I forcefully pointed out that the authors have not done any sequestration potential measurements but  simply completed a carbon balance analysis. I think the data presented can be used later for proper sequestration modeling. I still think the results provides an indicator for sequestration, so, the title as it is misleading and must be considered for amendment.

Reviewer 2 Report

Authors:

No need for any further edits. All the suggestion has been addressed.

The Reviewer
